# Influence of finite-time Lyapunov exponents on winter precipitation over Iberian Peninsula

Daniel Garaboa-Paz[1], Nieves Lorenzo[2], and Vicente Pérez-Muñuzuri[1]

[1]Group of Nonlinear Physics. Faculty of Physics. University of Santiago de Compostela. 15782 Santiago de Compostela, Spain.
[2]Ephyslab. Faculty of Sciences, Campus de Ourense, University of Vigo, 32004 Ourense, Spain.

*Correspondence to:* V. Pérez-Muñuzuri
(vicente.perez@cesga.es)

**Abstract.** Seasonal forecasts have improved during the last decades, mostly due to an increase of understanding of the coupled ocean-atmosphere dynamics, and the development of models able to predict the atmosphere variability. Correlations between different teleconnection patterns and severe weather in different parts of the world are constantly evolving and changing. This paper evaluates the connection between winter precipitation over the Iberian Peninsula and the large-scale tropospheric mixing over the eastern Atlantic ocean. Finite-time Lyapunov exponents (FTLE) have been calculated from 1979 to 2008 to evaluate this mixing. Our study suggests that significant negative correlations exist between summer FTLE anomalies and winter precipitation over Portugal and Spain. To understand the mechanisms behind this correlation, summer anomalies of the FTLE have also been correlated to other climatic variables as the sea surface temperature (SST), the sea level pressure (SLP) or the geopotential. The East Atlantic (EA) teleconnection index correlates with the summer FTLE anomalies confirming their role as a seasonal predictor for winter precipitation over the Iberian Peninsula.

## 1 Introduction

Seasonal forecast in mid-latitudes is still an open research field. However, the importance of this scale is relevant for different sectors such as agriculture (Howden et al., 2007; Meza et al., 2008), health (Thomson et al., 2008), energy (García-Morales and Dubus, 2007), or the financial sector (Meza et al., 2008). Most of Europe is located in the mid-latitude belt, where the changing nature of the atmosphere characteristics makes the seasonal climate forecast a difficult task. Thus, any tool to improve the forecast skill is potentially of great interest.

Two of the most important patterns that influence European climate variability are the North Atlantic Oscillation (NAO) and El Niño Southern Oscillation (ENSO) (Visbeck et al., 2003; Brönnimann, 2007). The indices of these large-scale climatic patterns are used as predictors for seasonal forecast over Europe (Lloyd-Hughes and Saunders, 2002). However, the relationship between NAO and ENSO and the European variability is nonstationary (Trigo et al., 2004); that is, the strength of the correlation between these two teleconnections and climate anomalies has changed over time. These patterns, though dominant on a large scale, often fail to provide forecast skill in specific regions. Precipitation predictability in Europe using NAO and ENSO as predictors is limited due to nonstationarity (Vicente-Serrano and López-Moreno, 2008; Rodríguez-Fonseca et al., 2016). One

way to improve the seasonal forecast would be to identify new predictors. Previous works have shown a possible link between the Iberian precipitation and other variables like summer sea surface temperature (SST) anomalies over the North Atlantic basin (Rodríguez-Fonseca and deCastro, 2002; Lorenzo et al., 2010; Hatzaki et al., 2015), other teleconnection patterns (deCastro et al., 2006; Casanueva et al., 2014) or the Euroasian snow cover in autumn (Brands et al., 2014). The storm track activity in mid-latitudes has been related to the occurrence of extreme events (Lehmann and Coumou, 2015). Changes in mid-latitude circulation are often directly linked to the occurrence of regional temperature and precipitation extremes (Marshall et al., 2001; Screen and Simmonds, 2014).

The atmosphere large-scale circulation causes mixing of air masses modifying the global moisture distribution, and therefore the rainfall patterns over the continents. One approach to characterize mixing and transport is by calculating Lagrangian trajectories of passive tracers in the atmosphere. Among the different statistics that can be calculated, finite-time Lyapunov exponents (FTLE) measure the separation of two trajectories over time from initially nearby starting points, i.e. the local mixing rates at a finite time (Shadden et al., 2005). FTLE have been used in atmospheric and oceanic studies to identify the presence of barriers to mixing in the atmosphere between the tropics and extratropics (Pierrehumbert and Yang, 1993), to study the zonal stratospheric jet (Beron-Vera et al., 2008), jet-streams (Tang et al., 2010), hurricanes (Rutherford et al., 2012), transient baroclinic eddies (von Hardenberg and Lunkeit, 2002), the polar vortex (Koh and Legras, 2002), atmospheric rivers (Garaboa-Paz et al., 2015), or the spread of plankton blooms (Huhn et al., 2012).

Our goal in this study is to characterize the rainfall patterns in the Iberian Peninsula as a function of the large-scale tropospheric mixing over the Atlantic ocean. To that end, we have calculated a climatology of FTLE using finite-difference approximation to the deformation gradient for the period $1979 - 2008$. The FTLE time series was then correlated with the precipitation over the Iberian Peninsula. Finally, we discuss the obtained results by considering their relationship to the main modes of circulation variability.

## 2 Methods

### 2.1 Data

The atmospheric transport has been studied using wind field data retrieved from the European Center for Medium-Range Weather Forecast reanalysis, ERA-Interim (Dee et al., 2011), for the $1979 - 2008$ period, with a horizontal spatial resolution of $0.7°$, a vertical resolution of 100 hPa and a temporal resolution of 6 hours.

A gridded dataset (IB02) was used for daily precipitation (mm) over the Iberian Peninsula from 1979 to 2008 (Ramos et al., 2014). This dataset covers the continental area of both Iberian countries on a high resolution ($0.2°$) grid. It is the combination of two different datasets; PT02 (Belo-Pereira et al., 2011) and SPAIN02 (Herrera et al., 2012).

Yearly anomalies of the SST, geopotential at 500 hPa, sea level pressure (SLP) and wind speed at 200 hPa and 850 hPa for the same period have been used to calculate the monthly/seasonal climate composites. The SST anomalies have been obtained from *The Extended Reconstructed Sea Surface Temperature* (ERSST) dataset which is a global monthly sea surface temperature dataset that constitutes part of the International Comprehensive Ocean-Atmosphere Dataset (ICOADS). It has

**Table 1.** Summary of the meteorological and climatological data sets.

| Acronym | | Web site |
| --- | --- | --- |
| ERA-Interim | Global atmospheric reanalysis | http://www.ecmwf.int/en/research/climate-reanalysis/era-interim |
| ERSST | Extended Reconstructed Sea Surface Temperature | https://www.ncdc.noaa.gov/data-access/marineocean-data |
| NCEP-NOAA | Earth System Research Laboratory | https://www.esrl.noaa.gov |
| CPC | Climate Prediction Center | http://www.cpc.noaa.gov/data/ |
| | Climate indices | https://www.esrl.noaa.gov/psd/data/climateindices/list/ |

been derived on a $2° \times 2°$ grid with spatial completeness enhanced using statistical methods. This monthly analysis begins in January 1854 continuing nowadays. The newest version of ERSST, version 4, is based on optimally tuned parameters using the latest datasets and improved analysis methods (Huang et al., 2015). The geopotential, SLP and wind speed anomalies have been obtained from the National Center of Environmental Prediction (NCEP) reanalysis with a spatial resolution of $2.5° \times 2.5°$

5 (Kalnay et al., 1996). The climatology used for the anomaly plots is for the $1981 - 2010$ period, considered to be the standard period for climatological studies of anomalies.

The most representative atmospheric patterns for the Northern Hemisphere were considered in order to analyze their influence on precipitation for the Iberian Peninsula and with the summer FTLEs. The teleconnection indices NAO, SCA (Scandinavia pattern), EA (East Atlantic pattern), EA/WR (East Atlantic/ West Russia pattern), POL (The Polar/ Eurasia pattern), SOI

10 (Southern Oscillation Index), PNA (Pacific-North American Pattern) and the atmospheric mode AO (Artic Oscillation) were obtained from the Climate Prediction Center (CPC) at NCEP at monthly time scales. Then, monthly indices were averaged for the seasons January-March (JFM), April-June (AMJ), July-September (JAS) and October-December (OND) from 1979 to 2008.

A summary of the climatological and meteorological web sites to download the data sets used in this study is given in

15 Table 1.

## 2.2 Finite-Time Lyapunov Exponents (FTLE)

In a longitude-latitude-pressure coordinate system $(\phi, \theta, P)$, the position of an air particle $\mathbf{r}(t) = (\phi(t), \theta(t), P(t))$ is calculated as $\dot{\mathbf{r}}(t) = \mathbf{v}(\mathbf{r}(t), t)$ and,

$$
\begin{aligned}
\dot{\phi}(t) &= \frac{u(\mathbf{r}(t), t)}{R \cos(\theta(t))} \\
\dot{\theta}(t) &= \frac{v(\mathbf{r}(t), t)}{R} \\
\dot{P}(t) &= w(\mathbf{r}(t), t)
\end{aligned} \tag{1}
$$

where, $u$, $v$ and $w$ are the eastward, northward and vertical wind components, respectively, and $R$ is the Earth's mean radius. A fine grid of particles with an initial separation of $0.35°$is uniformly distributed on the 850 hPa level to avoid turbulence effects from the boundary layer covering the domain $\mathbf{r}(t_0) = \{(\theta_0, \phi_0) \in [0, 360] \times [-85, 85]\}$ at time instant $t_0$. Then, 3D Lagrangian

simulations have been performed so that particle trajectories $\mathbf{r}(t;t_0,\mathbf{r_0})$ are computed integrating Eq. (1) using a 4-th order Runge-Kutta scheme with a fixed time step of $\Delta t = 1.5$ hours, and multilinear interpolation in time and space.

In order to characterize the atmospheric transport, we introduce the finite-time Lyapunov exponents (FTLE), that measure, at a given location, the maximum stretching rate of an infinitesimal fluid parcel over the time interval $[t_0, t_0 + \tau]$ starting at $\mathbf{r}(t_0;t_0,r_0) = \mathbf{r_0}$ and ending at $\mathbf{r}(t_0+\tau;t_0,\mathbf{r_0})$ (Shadden et al., 2005; Sadlo and Peikert, 2007). The integration time $\tau$ must be predefined and it has to be long enough to allow trajectories to explore the coherent structures present in the flow. The FTLE fields $\lambda$ are computed along the trajectories of Lagrangian tracers in the flow as (Peacock and Dabiri, 2010),

$$\lambda(\tau,t_0,\mathbf{r}_0) = \frac{1}{|\tau|} \log \sqrt{\mu_{max}(\tilde{\mathbf{C}}(\tau,t_0,\mathbf{r}_0))}, \tag{2}$$

where $\mu_{max}$ is the maximum eigenvalue of the pull-back Cauchy-Green deformation tensor $\tilde{\mathbf{C}}$ over a sphere (Haller and Beron-Vera, 2012) which does not take into account the deformation due to vertical movement. Repelling (attracting) coherent structures for $\tau > 0$ ($\tau < 0$) can be thought of as finite-time generalizations of the stable (unstable) manifolds of the system. These structures govern the stretching and folding mechanism that control flow mixing. To capture the main synoptic scales, an integration time of $\tau = 5$ days was selected which is about the mean length of the typical synoptic time scale in mid-latitudes. For larger time scales, observed coherent structures are smeared out, while for smaller $\tau$ values those structures are not well shaped, and multiple patterns arise.

Ridges in the FTLE field are used to estimate finite time invariant manifolds in the flow that separate dynamically different regions, and organize air masses transport. A positive time direction (forward FTLE) integration leads to identify lines of maximal divergence of air masses. In contrast, a negative time direction integration, leads to identify areas of maximal convergence (backward FTLE).

Following the steps explained above, a 30-years, $1979 - 2008$, time series of the FTLE field has been computed varying the initial time $t_0$ in fixed steps $\Delta t_0 = 6$ hours in order to release a new initial tracer grid. Each FTLE field obtained for each advection from $[t_0, t_0 + \tau]$ is an element of the time series $\lambda_i = \lambda(\tau,t_0 + i\Delta t_0, \mathbf{r_0})$.

FTLE anomalies were calculated from the FTLE median for the area $\mathbf{A}$ between $30°$W and $0°$W and between $25°$N and $65°$N for the period $1979 - 2008$; i.e. $\text{median}\,[<\lambda>_{\mathbf{A}}(t)]$. Seasonal composites (averages) of the anomalies (mean - total mean) of the SST, geopotential height and wind speed were obtained from NCEP for the same period. *"Total mean"* refers to the climatological period $1981 - 2010$ used for the anomaly plots by the Earth System Laboratory (NOAA). While *"mean"* corresponds to the mean of years with positive/negative (above/below the median) summer FTLE. Then, two time series (positive and negative phases) of these seasonal composites were calculated for years with positive/negative summer FTLE anomalies. Finally, maps shown below correspond to the time-averaged mean of both phases.

# 3   Results

We have studied the transport of air masses in terms of their FTLE from a climatological point of view. Figure 1(a) shows the FTLE for a given time at 850 hPa over the ocean. These structures reflect the large scale advection of air masses which are

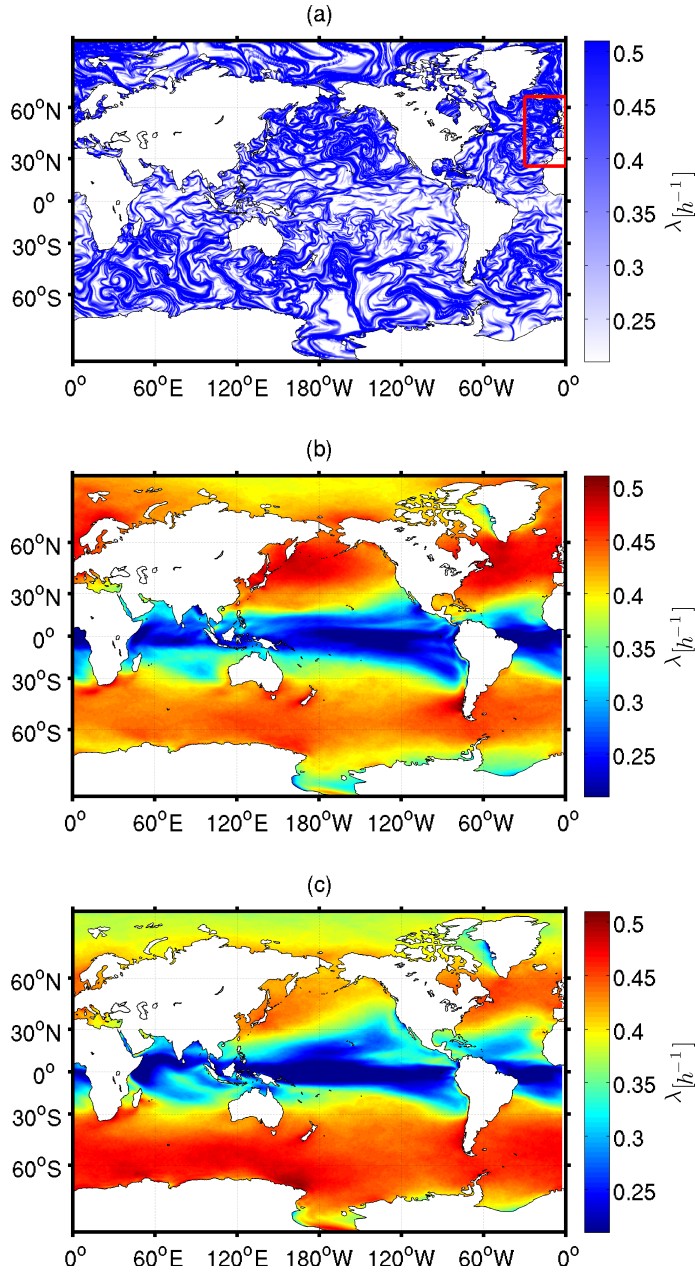

**Figure 1.** Finite-time Lyapunov exponents $\lambda$ (a). The box indicates the area where FTLE anomalies were calculated. Local maxima in the plot (darker colors) are potential repelling Lagrangian coherent structures. Mean FTLE climatology for periods January-March (JFM) (b) and July-September (JAS) (c) for the $1979 - 2008$ time series.

stretched and folded as wind transport them. The presence of ridges correspond to repelling manifolds where flow diverges. Time averaged FTLE maps for the $1979 - 2008$ period are shown in Figs. 1(b,c) for seasons January-March (JFM) and July-September (JAS), respectively. As it was expected, in both cases, three latitudinal bands can be clearly identified in coincidence with the large scale atmospheric circulation belts. For mid-latitudes, FTLE values are approximately two times higher than for the equatorial zone and large-scale mixing is generally stronger in winter (JFM) than in summer (JAS).

Westerly winds blowing across the Atlantic bring moist air into Europe. As the FTLE can be considered a measure of the large-scale tropospheric mixing (Garaboa-Paz et al., 2015), their role for JFM rainfall in the Iberian Peninsula has been considered. Figure 2 shows the Pearson's lag correlation between winter (JFM) precipitation in Iberian Peninsula and the anomalies of the FTLE for three different seasons through the period $1979 - 2008$.

The values obtained for the spring (Fig. 2(a)) and autumn (Fig. 2(c)) are not significant. However, the correlations obtained for summer (JAS) anomalies FTLE values (Fig. 2(b)) show large, spatially consistent and statistically significant negative correlations between the precipitation of the western half of the Iberian Peninsula and the FTLE. In other words, for larger negative summer FTLE anomalies (low values of the large-scale tropospheric mixing) in the eastern Atlantic ocean, next winter (JFM) precipitation over the western Iberian Peninsula will also be larger. This result suggests the possibility of using the FTLE as a tool to forecast the occurrence of significantly rainy winters in the considered area some months in advance.

The summer FTLE time series that show the largest correlation values with the winter (JFM) precipitation cover approximately the area **A** between $30°W$ and $0°W$ and between $25°N$ and $65°N$ (see the box in Fig. 1(a)). The size of this area was varied considering the entirely North Atlantic Ocean without modifying significantly the results shown here.

To gain insight into the observed correlations, positive and negative FTLE anomaly maps will be discussed in terms of the atmospheric circulation and the atmosphere-ocean interactions. Spatial patterns of atmospheric circulation and SST anomalies were calculated for the period $1979 - 2008$. Previous studies have shown that SST anomalies in certain areas of the subtropical North Atlantic are statistically related to winter (JFM) precipitation on the IPNA (Iberian Peninsula North Atlantic) region with a five-month lag (Rodríguez-Fonseca and deCastro, 2002).

Figure 3 shows the ratio between the positive (a) and negative (b) summer FTLE anomalies and the summer FTLE mean, respectively, and the summer SST anomalies calculated for the North Atlantic basin (c,d). Large values of the positive FTLE anomalies (a) correspond to a larger activity of the storm track and its displacement to the south during summer (JAS), in agreement with a weakening of the Azores anticyclone and negative SST anomalies for the middle North Atlantic Ocean, Fig. 3(c). On the contrary, low values of the negative FTLE anomalies (b) are associated to highs blocking the pass of cold fronts, thus reducing the large-scale atmospheric mixing. This pattern coincides with large positive SST anomalies observed during summer for the mid North Atlantic Ocean, Fig. 3(d). Both figures for SST anomalies (c,d) have a resemblance with the so-called *summer North Atlantic Horseshoe* (HS) SST pattern although displaced to the northwest. Previous works have found a relationship between summer SST anomalies for different areas of the North Atlantic basin and winter precipitation in Europe (Rodríguez-Fonseca and deCastro, 2002; Cassou et al., 2004). In our case, we hypothesize that changes in the large-scale tropospheric mixing during summer (JAS), measured in terms of the FTLE anomalies and also observed in the summer SST anomalies, are related to the winter (JFM) precipitation over Spain and Portugal. This result is in agreement with Lehmann

and Coumou (2015) and Dong et al. (2013) that observe a strong connection between changes in the mid-latitude circulation and extreme weather events.

Anomaly maps for the geopotential at 500 hPa, Fig. 4, show that positive summer FTLE anomalies correspond to the next winter (JFM) positive geopotential height anomalies occurring over western coast of Europe (a). On the contrary, summer (JAS) FTLE negative anomalies correspond to a next winter (JFM) trough, that is a region with relatively lower geopotential heights, occurring over the western coast of Europe and in particular over the western Iberian Peninsula (b). These troughs are associated to cloudy conditions and precipitation. A similar pattern is observed for the SLP anomaly patterns (not shown here).

In addition, winter (JFM) wind speed anomalies for 200 hPa and 850 hPa are shown in Fig. 5. Weaker winds are observed at latitudes between 40°N and 50°N for the anomalies at 200 hPa associated with positive FTLE anomalies (a). This can be related to a weaker than normal jet stream. As the jet stream defines the storm track, a weaker jet stream means that lows traveling across the Atlantic do not reach the Iberian Peninsula with the expected frequency. Moreover, taking into account that winds are larger at latitudes $\approx 30°$N (Fig. 5(a)) and that geopotential anomalies show high pressures over the NE Atlantic and low pressures located at NW Atlantic (Fig. 4(a)) we can hypothesize that the meridional variability of the jet stream increases under these circumstances. Figure 5(c), also for positive summer (JAS) FTLE anomalies, shows low wind speed values at 850 hPa related to a reduced transport of moisture and cold fronts with less precipitation in the path connecting the Caribbean area and the Iberian Peninsula. Note that easterlies are displaced northward. Figures 5(b,d) show the inverse situation for negative summer FTLE anomalies. In Fig. 5(b) there are no significant anomalies in the storm track. Therefore, the jet stream is at its usual location and the Iberian Peninsula is affected by low pressure systems. Note in Fig. 5(d) that for mid-latitudes and 850 hPa, the connection between the Caribbean area and the Iberian Peninsula is reinforced, meaning that rainfall may be stronger associated with the enhancement of the poleward transport of moisture. As well, weaker easterlies are displaced to lower latitudes reinforcing westerlies between the Caribbean area and the Iberian Peninsula.

Finally, summer (JAS) anomalies of the FTLE and winter (JFM) precipitation in the Iberian Peninsula have been correlated to different teleconnection patterns. Table 2 summarizes the main results. Note that summer anomalies of the FTLE and summer and autumn EA index correlate (positive/negative, respectively) with a significance greater than 95%, which also correlates with winter (JFM) rainfall. EA index is the second mode of inter-annual variability of the tropospheric circulation of the North Atlantic. Previous works have found that in southern Europe the EA pattern is at least as important as the NAO for explaining inter-annual variations of sensible climate variables such as air temperatures, sea-surface temperatures, precipitation or wind (Serrano et al., 1999; Saenz et al., 2001; Comas-Bru and McDermott, 2013; Casanueva et al., 2014). Besides, some recent studies suggest that the EA pattern may play a role in positioning the primary North Atlantic storm track (Seierstad et al., 2007; Woollings et al., 2010). Note as well in Table 2 that the autumn SCA pattern also correlates negatively with the summer FTLE anomalies although less significant than EA (90% significant), and this pattern influence the winter (JFM) precipitation. The Scandinavia pattern (SCA) consists of a primary circulation center over Scandinavia, with weaker centers of opposite sign over western Europe and eastern Russia/ western Mongolia. The Scandinavia pattern has been previously referred to as the Eurasia-1 pattern by Barnston and Livezey (1987) and other studies also have shown its influence on the Iberian Peninsula precipitation (deCastro et al., 2006; Casanueva et al., 2014).

**Table 2.** Seasonal correlations between summer (JAS) FTLE anomalies, winter (JFM) rainfall and different teleconnection indices; the North Atlantic Oscillation (NAO), the East Atlantic Pattern (EA), the East Atlantic/Western Russia (EA/WR), the Scandinavian Pattern (SCA), the Pacific/North American teleconnection pattern (PNA), the Southern Oscillation Index (SOI), and the Arctic Oscillation (AO). Only significant values are shown. (*) and (**) stand for significances larger than 95% and 90%, respectively. $(+1, 0, -1)$ correspond to the preceding, same and following seasons, respectively.

| Summer (JAS) FTLE | | | | | | | |
|---|---|---|---|---|---|---|---|
| Season/index | NAO | EA | EA/WR | SCA | PNA | SOI | AO |
| JFM (+1) | | | | | | -0,24* | |
| AMJ (+1) | 0,34** | | | | 0,23* | | |
| JAS (0) | | 0,46** | | -0,26* | 0,32** | | 0,25* |
| OND (0) | | -0,54** | | -0,27* | -0,30** | | 0,29* |
| Winter (JFM) Rainfall | | | | | | | |
| Season/index | NAO | EA | EA/WR | SCA | PNA | SOI | AO |
| JFM (0) | -0,58** | | -0,32** | 0,69** | | 0,37** | -0,52** |
| AMJ (-1) | | | | 0,27* | | | |
| JAS (-1) | | -0,38** | | | -0,25* | | |
| OND (-1) | | 0,46** | | 0,36** | | | -0,48** |

## 4 Conclusions

In the present study, the connection between winter (JFM) precipitation over the Iberian Peninsula and the finite-time Lyapunov exponents (FTLE) calculated over the eastern Atlantic ocean has been investigated. A significant negative correlation was observed between the summer (JAS) FTLE anomalies and winter (JFM) precipitation over the western Iberian Peninsula. The results suggest the possibility of predicting the occurrence of rainy winters using these exponents several months in advance.

Summer anomalies of the FTLE have also been correlated to other circulation and temperature patterns as SST, SLP or the geopotential. In all cases, negative summer (JAS) FTLE anomalies correspond to well known patterns of precipitation over the western Iberian Peninsula. Low values of tropospheric mixing (negative FTLE anomalies) during summer (JAS) correspond to positive SST anomalies and highs blocking the passage of fronts over the western Europe. However, wind speed, SLP and geopotential anomalies during next winter show the opposite relationship as lows may reach the Iberian Peninsula, in agreement with the negative correlation observed between summer FTLE and winter (JFM) precipitation. Our results are in agreement with Cayan (1992) who showed a strong dependence between heat flux, SST anomalies and the SLP modes on large spatial scales. The heat flux anomalies, derived from bulk formulations, exhibit large-scale patterns of variability which are related to patterns of sea level pressure (SLP) variability and also to patterns of SST anomalies. In our case, we showed that FTLE anomalies correspond to patterns of SST and SLP variability. In our opinion, large-scale tropospheric mixing drives summer SST anomalies that lead to changes in the next seasons storm tracks, and consequently changes in the location of action centers, Fig. 4 (low and high pressures centers). Finally, the relationship with some teleconnection patterns of the Northern Hemisphere

was shown in Table 2. Once more, summer FTLE anomalies correlate with summer and next autumn EA indices influence winter rainfall patterns of the Iberian Peninsula.

*Acknowledgements.* ERA-Interim data were provided by ECMWF. This work was financially supported by Ministerio de Economía y Competitividad and Xunta de Galicia (CGL2013-45932-R, GPC2015/014), and contributions by the COST Action MP1305 and CRETUS Strategic Partnership (AGRUP2015/02). All these programmes are co-funded by ERDF (EU). Computational part of this work was done in the Supercomputing Center of Galicia, CESGA. We acknowledge fruitful discussions with J. Eiras-Barca.

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

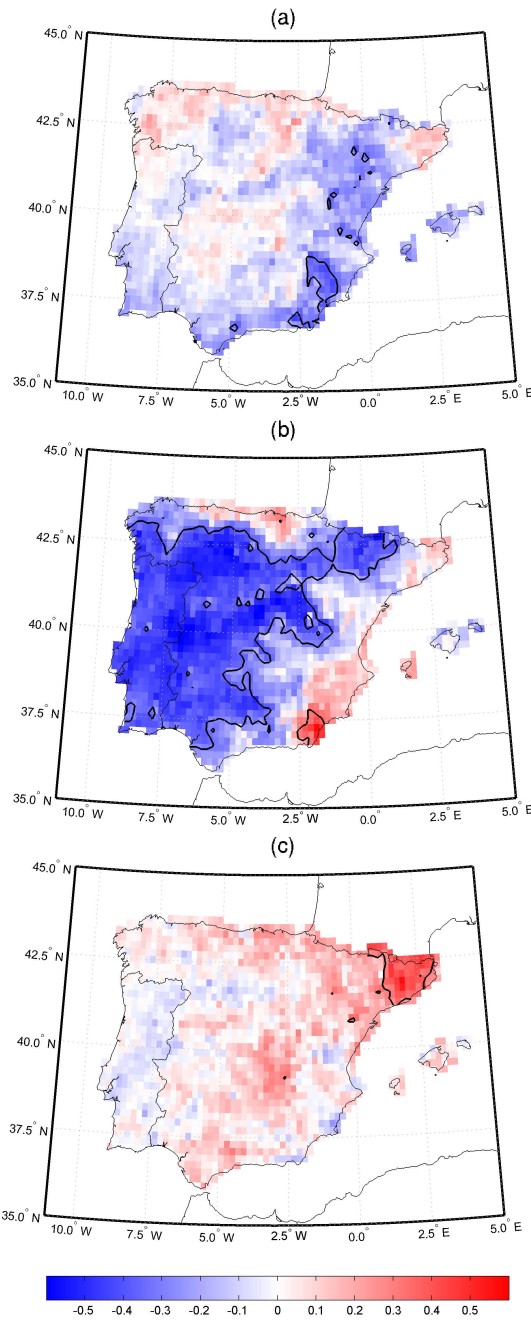

**Figure 2.** Correlation map between winter (JFM) precipitation of the Iberian Peninsula and the FTLE anomalies for the Eastern Atlantic region ($[30°W, 0°W] \times [25°N, 65°N]$) for the three previous seasons; a) spring (AMJ), b) summer (JAS) and c) autumn (OND). The black lines mark the significant correlation at 95% according to the Student's t test.

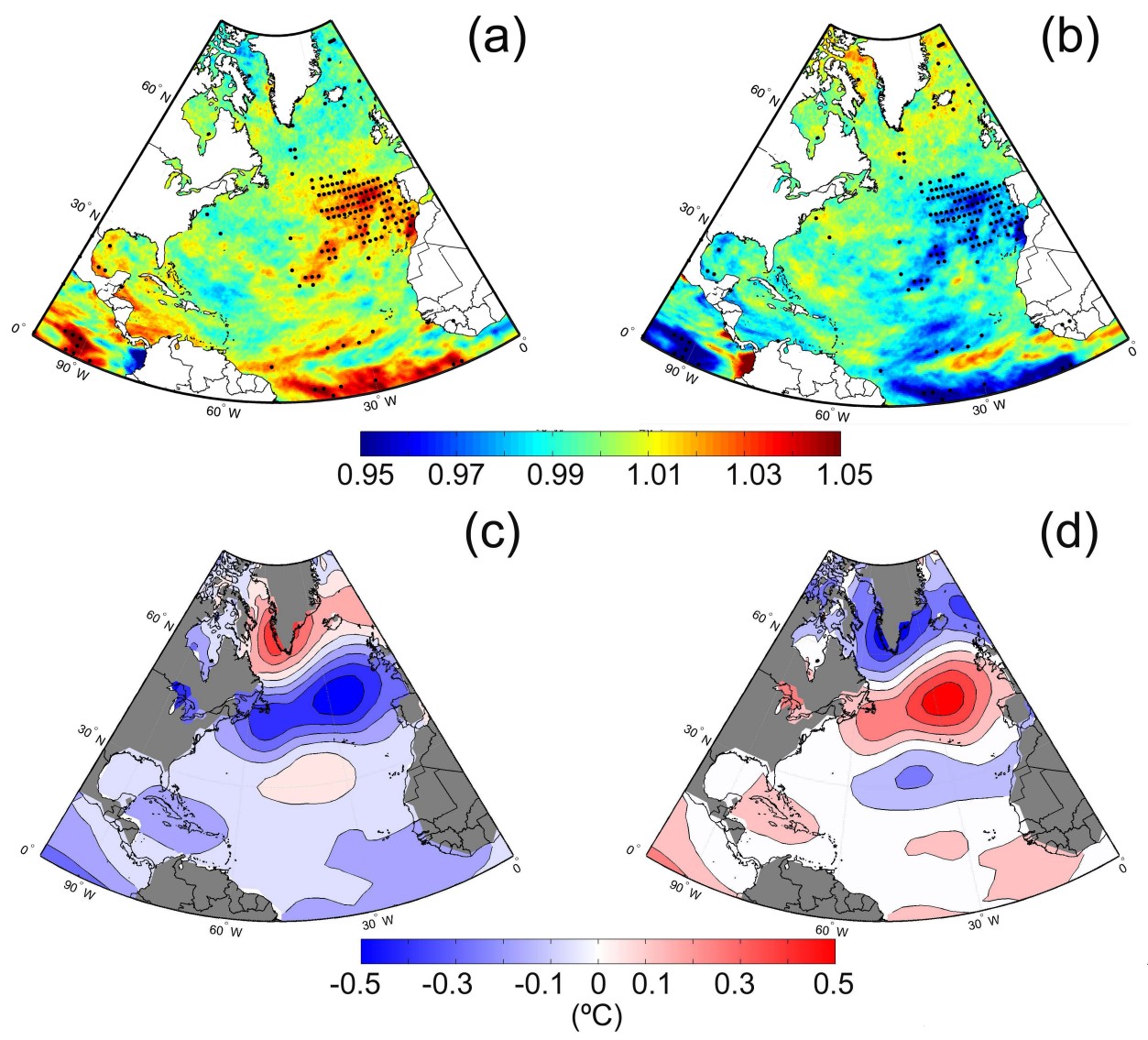

**Figure 3.** Ratio between the positive (a) and negative (b) summer (JAS) FTLE anomalies and the summer FTLE mean. Summer SST anomalies in the Atlantic ocean for years with positive (c) and negative (d) summer FTLE anomalies, respectively. Pointed areas in panels (a) and (b) display the significant differences between both phases obtained from a two-sided Wilcoxon ranksum test.

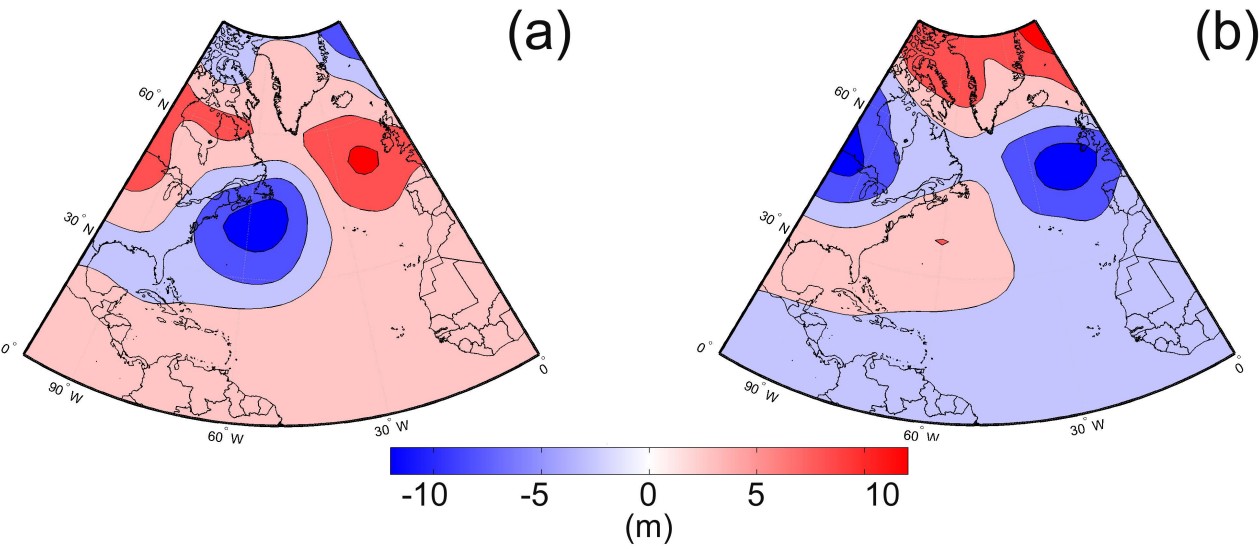

**Figure 4.** Geopotential at 500 hPa anomalies in the Atlantic for winter. (a) and (b) correspond to years with positive and negative summer FTLE anomalies, respectively.

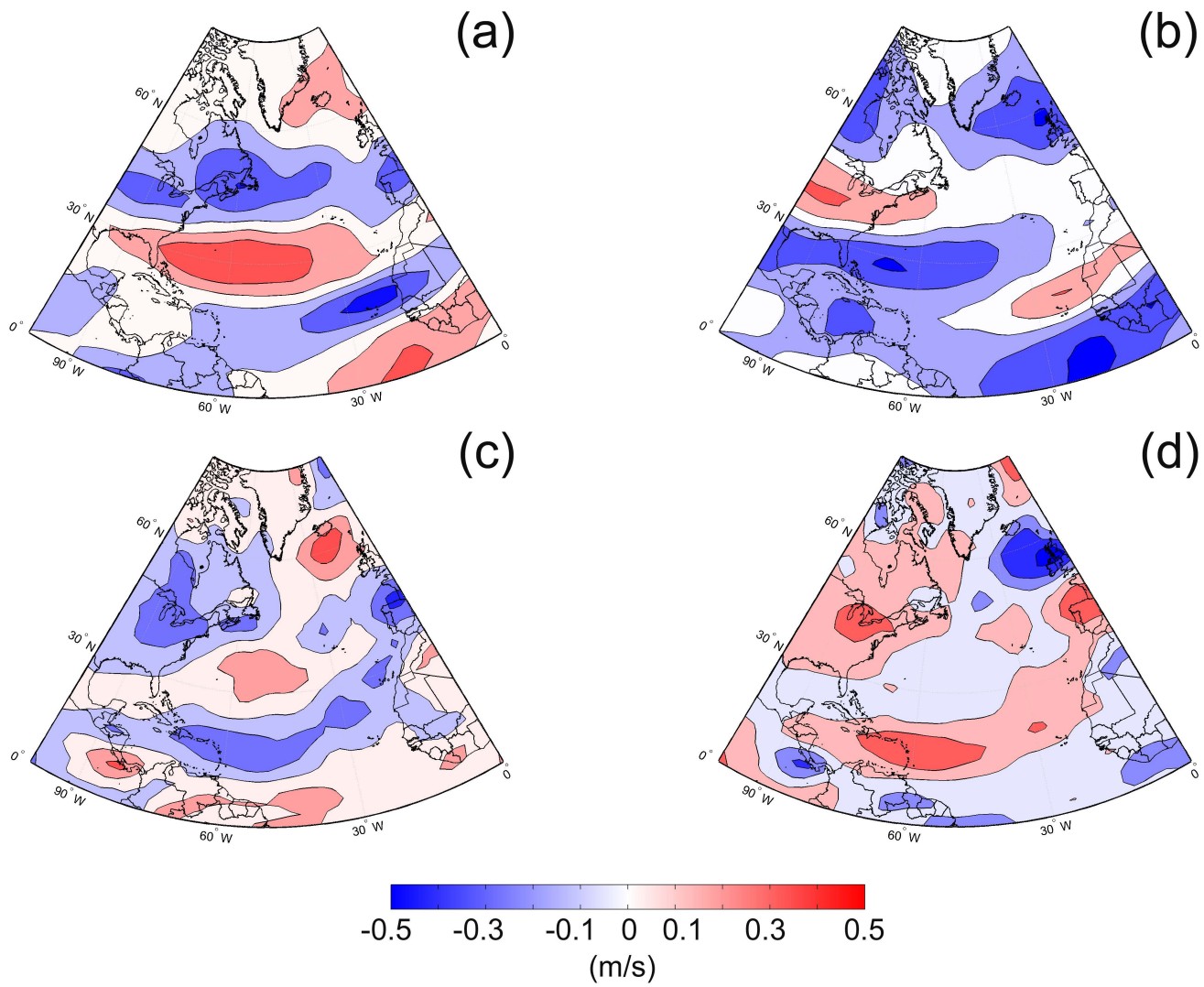

**Figure 5.** Wind speed anomalies in the Atlantic for winter (JFM). (a/b) and (c/d) correspond to years with positive/negative summer (JAS) FTLE anomalies. Upper and lower maps correspond to 200 hPa and 850 hPa levels, respectively.