# Peer review of "Influence of finite-time Lyapunov exponents on winter precipitation over Iberian Peninsula"

_Nonlinear Processes in Geophysics, 2016_

## Referee Comment (RC1) · Anonymous Referee #1 · 23 Jan 2017

The manuscript calculates the local atmospheric mixing using finite time Lyapunov exponents and finds that summertime changes in this field are correlated with winter changes in precipitation over the Iberian peninsula. Thus, they suggest this field could be used as a predictor of precipitation in this region.

The topic is very relevant and important because seasonal prediction is still a very active area of research. Thus, any advance in this area will be welcome. However, I am not convinced about the interpretation and the mechanism that connects mixing in summer with rainfall in winter. As the authors mention other works have already found a connection between summertime north Atlantic SST and winter rainfall in Europe. Here, the authors try to explain their results expanding on those based on composites.

But, I could not follow the reasoning of the authors: what is the causality link between changes in mixing in summer and rainfall in winter? Through changes in the SST? Does the SST somehow force a certain teleconnection pattern during winter, which in turn changes rainfall?

Also, changes in mixing are very small, close to 3%... are they significant? Moreover, are the SST, geopotential height, and wind anomalies statistically significant for positive and negative cases of FTLE? Authors need to quantify this, maybe through a test of differences between positive/negative years and neutral years.

Lines 285-287 and 292-295 imply that climatological winds are westerly between the Caribbean and the Iberian Peninsula, but south of 25N the easterly trade winds dominate. Thus, these sentences are incorrect.

Other comments:

-the sentence "less wind than normal" does not sound right. Please change to "weaker winds" or similar.

- why did you chose the 850 hPa level, which is usually just above the boundary layer? Does mixing change significantly depending on the level? If the deformation due to vertical movement is not taken into consideration, shouldn't you pick a level where the atmosphere tends to behave in 2D? Maybe upper levels?

- why did you chose tau=5 days? Is it to capture the mixing due to synoptic variability? Have you performed a sensitivity test by changing tau within 1 or 2 days?

---

## Referee Comment (RC2) · R. V. Donner (Referee) · 27 Jan 2017

This manuscript proposes an interesting idea of relating dynamical indicators for atmospheric mixing with regional precipitation. By performing a correlation analysis between seasonally averaged summer-time finite-time Lyapunov exponents, winter precipitation and several teleconnection indices, they establish statistical linkages between these variables, which could be further associated with certain climatic processes and physical mechanisms.

While the manuscript contains some potentially relevant findings, some aspects have left me slightly confused and need to be clarified in a revised paper.

[Figure]

First, the title refers to "seasonal predictability" of winter precipitation, as opposed to "seasonal prediction". This might be a subtle difference, but the readership of Non-linear Processes in Geophysics might wish to distinguish between both aspects. The problem is that I did not really find the "seasonal predictability" (as a nonlinear dynamic characteristic) of the winter precipitation records being quantified (rather, one could argue that the FTLE fields discussed provide a means to quantify the spatio-temporally local predictability of atmospheric flow). I am not convinced that at the considered level of seasonal aggregates, it is even possible to quantify the predictabilits of seasonal precipitation sums, given the available time span of observations. On the other hand, I also did not find the aspect of "prediction" being specifically addressed at all (which would essentially mean building a regression(?) model for seasonal precipitation sums based on covariates identified by the performed correlation analysis.

Second, it is appreciated that the authors use dynamical characteristics of the atmospheric circulation to establish a kind of "climatology" in terms of statistical relationships with teleconnection indices. This is most valuable for obtaining a process-based understanding of the observations made. However, it is not clear to me at all why the authors define their four seasons as "JFM", "AMJ", "JAS" and "OND" instead of using the classical - and climatologically well motivated - definitions "DJF", "MAM", "JJA" and "SON". The problem is that when using the terms "summer" and "winter" in the paper, the corresponding definitions do not match what is usually understood by climatologists when using these terms. This makes it hard to establish clear relationships between the findings of the present paper and those of previous works. I strongly recommend revising the results by sticking to the established definitions of seasons.

Third, I recommend giving precise definitions/explanations of how the different types of anomalies used in the paper are calculated. In some cases, this is not obvious from the text and makes evaluating the obtained results quite hard.

Fourth, atmospheric circulation is highly dynamic and involves a multiplicity (actually, a continuum) of spatial and temporal scales. I think that it can be justified to restrict

the attention within the present work to a single atmospheric layer (850 hPa pressure level) and a constant integration time (5 days; this information should be given in the main text instead of a figure caption), but the motivation of both specific choices should be made transparent. I wonder how much the obtained FTLE fields and established statistical relations may depend on the pressure level at which the tracers are initiated. Moreover, how much can we actually learn from time-averaged FTLEs given that Lagrangian coherent structures, hyperbolic trajectories and related objects embedded in the atmospheric flow are not stationary over the seasonal time scales considered in this work? I am willing to accept that the seasonally averaged FTLE fields still provide useful and interpretable information, but what is beyond the mean? For example, does the variance of FTLEs show similar and possibly relevant spatio-temporal patterns? I think that what the authors present is an interesting starting point, but much more could (and should) be done in this regard.

Finally, the authors just report a relationship between summer mixing and winter precipitation, but I do not find information describing a corresponding physical linkage connecting both seasons. At least some speculations about corresponding mechanisms should be given.

Specific comments:

* p.1, l.3: Teleconnection patterns and severe weather (events) have not just evolved during the last years, but are constantly changing.

* When working with wind data, please specific if you consider just the wind speed or the full vector field.

* p.3, l.10: "the significance of this coefficient was assessed to be greater than 95%" is a quite awkward formulation

* p.4, l.19: What do you mean by "lead-lag correlation"?

* p.4, l.20: What is the "North Atlantic East Ocean"?

* p.5, l.1: What is the "JPNA region"?

* p.5, l.33 and below: Please be specific in whether correlations are positive or negative.

* Tab. 1: use capital letters for indicating calendar months

* p.6, l.9: SCA is not the third leading mode of WINTER SLP variability, but can be computed for all seasons (as every teleconnection index).

In addition, the English could be further polished here and there, especially regarding the proper use of articles and (in just a few cases) the consistency of tenses.

———————————————————

---

## Author Comment (AC1) · 21 Feb 2017

We would like to thank the Referee for his/her valuable comments and critics that we tried to take into account in the revised version of the manuscript. Hopefully, all the major and minor corrections pointed out by the reviewer have been corrected now. A detailed answer follows below. We provide replies to the reviewer' comments in bold. As well, corrections included in the manuscript are marked in red.

Answer to Referee 1

The manuscript calculates the local atmospheric mixing using finite time Lyapunov exponents and finds that summertime changes in this field are correlated with winter

changes in precipitation over the Iberian Peninsula. Thus, they suggest this field could be used as a predictor of precipitation in this region.

The topic is very relevant and important because seasonal prediction is still a very active area of research. Thus, any advance in this area will be welcome. However, I am not convinced about the interpretation and the mechanism that connects mixing in summer with rainfall in winter. As the authors mention other works have already found a connection between summertime north Atlantic SST and winter rainfall in Europe. Here, the authors try to explain their results expanding on those based on composites.

But, I could not follow the reasoning of the authors: what is the causality link between changes in mixing in summer and rainfall in winter? Through changes in the SST? Does the SST somehow force a certain teleconnection pattern during winter, which in turn changes rainfall?

The interaction between the ocean and atmosphere is complex. Heat and momentum flux at the interface modify currents and winds near the surface. On the other hand, Cayan showed that vast regions of the middle-latitude ocean surface temperature variability is forced by the atmospheric variations. He showed a strong dependence between heat flux, SST anomalies and the SLP modes on spatial scales that often span major portions of the North Atlantic. The heat flux anomalies, derived from bulk formulations, exhibit large-scale patterns of variability which are related to patterns of sea level pressure (SLP) variability and also to patterns of SST anomalies.

In our case, we showed that FTLE anomalies correspond to patterns of SST and SLP variability. In our opinion, large-scale tropospheric mixing drives summer SST anomalies that lead to changes in the next seasons storm tracks, and consequently changes in the location of action centers (low and high pressures centers).

• Cayan, D.R. Latent and sensible heat flux anomalies over the northern oceans: Driving the sea surface temperature. J. of Phys. Ocean. 22, 859-881, 1992.

Also, changes in mixing are very small, close to 3%... are they significant? Moreover, are the SST, geopotential height, and wind anomalies statistically significant for positive and negative cases of FTLE? Authors need to quantify this, maybe through a test of differences between positive/negative years and neutral years.

Although the changes in the values of positive phase and negative phase of FTLEs are small, the Wilcoxon rank sum test shows that these differences between positive and negative phase of FTLEs are significant. The same is found between the differences of the positive and negative anomalies of the other variables.

Lines 285-287 and 292-295 imply that climatological winds are westerly between the Caribbean and the Iberian Peninsula, but south of 25N the easterly trade winds dominate. Thus, these sentences are incorrect.

We agree with the reviewer, perhaps in the manuscript we describe Figures 5c and 5d in such a way that they were misunderstood. We have rewritten the corresponding paragraph in the manuscript.

For some particular years, weaken easterlies are displaced to lower latitudes reinforcing westerlies between the Caribbean area and the Iberian Peninsula, meaning that rainfall may be stronger associated with the enhancement of the poleward transport of moisture (see for example, figure for January 2001, a month corresponding to a particularly rainy winter). This situation is a particular case of Fig.5d. However, for other years, easterly winds predominate as they are displaced towards North. Low wind speed values are then observed in the trajectory between Caribbean area and the Iberian Peninsula related to a less frequent arrival of cold fronts and their associated baroclinic structures (see for example, figure for January 2005, a month corresponding to a particularly dry winter). This situation is a particular case of Fig.5c.

The sentence "less wind than normal" does not sound right. Please change to "weaker winds" or similar.

The reviewer is right. The sentence has been modified.

Why did you choose the 850 hPa level, which is usually just above the boundary layer (PBL)?

We want to focus on the troposphere, but at the same time we wanted to avoid the atmospheric events close to the surface within the PBL. We are interested in the large-scale tropospheric mixing. To that end, we start the advection at the intermediate level of 850hPa so the observed coherent structures are not perturbed by turbulence effects coming from the PBL.

Does mixing change significantly depending on the level?

We perform integrations at different levels 850 hPa, 500 hPa and 300 hPa. In some cases, the main synoptic coherent structures remain qualitatively the same (see for example 3D simulations in Garaboa-Paz et al, 2015) at different levels; the strength of the FTLE ridges diminishes as pressure decreases. We notice that some structures become weaker and the integration time should be modified to capture these structures. However, in some other cases, due to the different flow dynamics occurring at different pressure levels, the observed coherent structures do not coincide. However, we expect the climatological means and anomalies calculated at different levels to behave similarly.

If the deformation due to vertical movement is not taken into consideration, shouldn't you pick a level where the atmosphere tends to behave in 2D? Maybe upper levels?

The particles are advected in 3D, but, only the deformation due to 2D horizontal movement is considered. We want to focus on the horizontal spatial deformation instead of vertical deformation.

The vertical-horizontal scales are completely different in the atmosphere, so considering the deformation due to vertical movement will lead to define a 3x3 Cauchy Green tensor. The eigenvalues of this matrix only take into account the relative deformation of

an ellipsoid respect to their initial conditions, without distinction between horizontal or vertical movement. If the vertical deformations have the same weight than horizontal deformation, this could lead to mask the FTLE values.

Why did you chose tau=5 days? Is it to capture the mixing due to synoptic variability? Have you performed a sensitivity test by changing tau within 1 or 2 days?

Yes, the reviewer is right. We want to capture the main synoptic scales. Five days is about the mean length of the typical synoptic time scale in mid-latitudes. For larger time scales, observed coherent structures are smeared out, while for smaller tau values those structures are not well shaped, and multiple patterns arise.

Please also note the supplement to this comment:
http://www.nonlin-processes-geophys-discuss.net/npg-2016-79/npg-2016-79-AC1-supplement.pdf
* * *
[Figure]

**Fig. 1.** Examples for wind behavior

**Supplement:**

[revised manuscript text omitted]

---

## Author Comment (AC2) · 21 Feb 2017

We would like to thank the Referee for his/her valuable comments and critics that we tried to take into account in the revised version of the manuscript. Hopefully, all the major and minor corrections pointed out by the reviewer have been corrected now. A detailed answer follows below. We provide replies to the reviewer' comments in bold. As well, corrections included in the manuscript are marked in red.

Answer to Referee 2

First, the title refers to "seasonal predictability" of winter precipitation, as opposed to "seasonal prediction". This might be a subtle difference, but the readership of Non-

linear Processes in Geophysics might wish to distinguish between both aspects. The problem is that I did not really find the "seasonal predictability" (as a nonlinear dynamic characteristic) of the winter precipitation records being quantified (rather, one could argue that the FTLE fields discussed provide a means to quantify the spatio-temporally local predictability of atmospheric flow). I am not convinced that at the considered level of seasonal aggregates, it is even possible to quantify the predictability of seasonal precipitation sums, given the available time span of observations. I also did not find the aspect of "prediction" being specifically addressed at all (which would essentially mean building a regression(?) model for seasonal precipitation sums based on covariates identified by the performed correlation analysis.

We agree with the reviewer, may be the title of the paper is not clear. In this paper, we do not intend to construct a prediction model of winter precipitation based on FTLEs. Our goal is to analyze the potential of FTLEs to see the possibility of considering them as a seasonal prediction tool. So, in order to avoid misunderstandings, we prefer to change the title of the manuscript,

Influence of Finite-time Lyapunov exponents on winter precipitation over Iberian Peninsula

Second, it is appreciated that the authors use dynamical characteristics of the atmospheric circulation to establish a kind of "climatology" in terms of statistical relationships with teleconnection indices. This is most valuable for obtaining a process-based understanding of the observations made. However, it is not clear to me at all why the authors define their four seasons as "JFM", "AMJ", "JAS" and "OND" instead of using the classical - and climatologically well motivated - definitions "DJF", "MAM", "JJA" and "SON". The problem is that when using the terms "summer" and "winter" in the paper, the corresponding definitions do not match what is usually understood by climatologists when using these terms. This makes it hard to establish clear relationships between the findings of the present paper and those of previous works. I strongly recommend revising the results by sticking to the established definitions of seasons.

To our viewpoint there is not a standard definition of seasons. It depends on the scope of the study. Moreover, there exists a large amount of literature where the seasonal periods are defined as we did. Some examples where winter is assimilated to (JFM) are,

- Gastineau G. D'Andrea F. Frankignoul C. (2013) Atmospheric response to the North Atlantic Ocean variability on seasonal to decadal time scales. Clim Dyn (2013) 40:2311–2330 DOI 10.1007/s00382-012-1333-0 - Picado A, Alvarez I, Vaz N, Varela R, Gómez-Gesteira M, Dias JM. 2014. Assessment of chlorophyll variability along the northwestern coast of Iberian Peninsula. J. Sea Res. 93: 2–11, doi: 10.1016/j.seares.2014.01.008. - Bamzai A. S. (2003) Relationship between snow cover variability and Arctic oscillation index on a hierarchy of time scales Int. J. Climatol. 23: 131–142 (2003) doi: 10.1002/joc.854

Third, I recommend giving precise definitions/explanations of how the different types of anomalies used in the paper are calculated. In some cases, this is not obvious from the text and makes evaluating the obtained results quite hard.

We agree with the reviewer, so we add some explanations at the end of the Methods section 2.1. Thanks to point us this.

Seasonal composites (averages) of the anomalies (mean - total mean) of SST, geopotential height and wind speed were obtained from the page (https://www.esrl.noaa.gov/psd/cgi-bin/data/composites) for the period 1979-2008. Total mean makes reference to the climatological mean in this case 1981-2010.

Then, two time series (positive and negative phases) of these seasonal composites were calculated for years with positive/negative summer FTLE anomalies. Finally, Figs.(3-5) show the time-averaged mean of both phases.

Fourth, atmospheric circulation is highly dynamic and involves a multiplicity (actually, a continuum) of spatial and temporal scales. I think that it can be justified to restrict

the attention within the present work to a single atmospheric layer (850 hPa pressure level) and a constant integration time (5 days; this information should be given in the main text instead of a figure caption), but the motivation of both specific choices should be made transparent. I wonder how much the obtained FTLE fields and established statistical relations may depend on the pressure level at which the tracers are initiated.

We agree with the referee with this insight. We move the information given in the caption to the main text and add some details to clarify this description.

We want to focus on the troposphere, but at the same time we wanted to avoid the atmospheric events close to the surface within the PBL. We are interested in the large-scale tropospheric mixing. To that end, we start the advection at the intermediate level of 850hPa so the observed coherent structures are not perturbed by turbulence effects coming from the PBL.

With respect to the integration time, five days is about the mean length of the typical synoptic time scale in mid-latitudes. For larger time scales, observed coherent structures are smeared out, while for smaller tau values those structures are not well shaped, and multiple patterns arise.

Moreover, how much can we actually learn from time-averaged FTLEs given that Lagrangian coherent structures (LCS), hyperbolic trajectories and related objects embedded in the atmospheric flow are not stationary over the seasonal time scales considered in this work?

Lagrangian coherent structures correspond to flow regions where mixing is larger than the average for a period of tau days. The activity, intensity and presence of these coherent structures in the atmosphere are highly influenced by the atmospheric flow. Their position and shape evolves with time as the flow does. The evolution rate depends on how fast the flow changes as it happens with the rest of atmospheric structures. The FTLE have been widely used in oceanography and meteorology to describe transport phenomena. In our opinion, trying to understand the atmospheric dynamics over the

seasonal time scales with the FTLE is equivalent to do it with the SST, SLP, etc. The FTLE describe the amount of large-scale tropospheric mixing available to perturb the SST.

I am willing to accept that the seasonally averaged FTLE fields still provide useful and interpretable information, but what is beyond the mean? For example, does the variance of FTLEs show similar and possibly relevant spatio-temporal patterns? I think that what the authors present is an interesting starting point, but much more could (and should) be done in this regard.

This is an interesting question and we acknowledge the reviewer to point us this idea.

As a first approach, and in a different context, we have performed some simulations to study the variability of the FTLE in terms of the intra-annual (standard deviation of the monthly means for the whole period) and inter-annual (standard deviation of the annual means) variabilities (see pictures below).

These results highlight El Niño Southern Oscillation, the storm track or the Intertropical Convergence Zone among other large-scale structures.

A time-series consisting of the variance of the FTLEs within the region studied in this manuscript, or the intra or inter-season variabilities of the FTLE could also be used to correlate them with the winter precipitation. However, in our opinion we believe that this study deserves further work and it is beyond the scope of this manuscript.

Finally, the authors just report a relationship between summer mixing and winter precipitation, but I do not find information describing a corresponding physical linkage connecting both seasons. At least some speculations about corresponding mechanisms should be given.

In the Introduction of our manuscript several references were given to previous works that established a possible link between the Iberian precipitation and other variables like summer Sea Surface Temperature (SST) anomalies over the north Atlantic basin

(Rodríguez-Fonseca and deCastro, 2002; Lorenzo et al., 2010; Hatzaki et al., 2015), other teleconnection patterns (deCastro et al., 2006; Casanueva et al., 2014) or the Euroasian snow cover in autumn (Brands et al., 2014).

Concerning the mechanisms linking the atmospheric variability with precipitation, we believe that it happens via changes in the SST. The interaction between the ocean and atmosphere is complex. Heat and momentum flux at the interface modify currents and winds near the surface. Cayan showed that vast regions of the middle-latitude ocean surface temperature variability are forced by the atmospheric variations. He showed a strong dependence between heat flux, SST anomalies and the SLP modes on spatial scales that often span major portions of the North Atlantic. The heat flux anomalies, derived from bulk formulations, exhibit large-scale patterns of variability which are related to patterns of sea level pressure (SLP) variability and also to patterns of SST anomalies. In our case, we showed that FTLE anomalies also correspond to patterns of SST and SLP variability. In our opinion, large-scale tropospheric mixing drives summer SST anomalies that lead to changes in the next seasons storm tracks, and consequently changes in the location of action centers (low and high pressures centers).

- Cayan, D.R. Latent and sensible heat flux anomalies over the northern oceans: Driving the sea surface temperature. J. of Phys. Ocean. 22, 859-881, 1992.

Specific comments:

p.1, l.3: Teleconnection patterns and severe weather (events) have not just evolved during the last years, but are constantly changing.

We have modified the Abstract.

When working with wind data, please specific if you consider just the wind speed or the full vector field.

Yes, you are right. We have modified the text. We used the scalar Wind Speed.

p.3, l.10: "the significance of this coefficient was assessed to be greater than 95%" is a quite awkward formulation

We agree with the reviewer. We have modified the text at the end of Section 2.1 as, The Pearson correlation coefficient and the Student's t test were used to identify the statistical significance of the correlations between the anomalies of the FTLE and precipitation.

p.4, l.19: What do you mean by "lead-lag correlation"?

Lead-lag correlation, describes the situation where one (leading) variable cross-correlated with the values of another (lagging) variable. But, probably you are right and we do not use the correct term, so we changed it to "lag correlation".

p.4, l.20: What is the "North Atlantic East Ocean"?

We have modified the manuscript in page 4 at the end of the Methods Section as follow, FTLE anomalies were calculated from the FTLE median for the area between 30°W and 0°W and between 25°N and 65°N for the period 1979-2008.

Later on, within the Results section,

Figure 2 shows the lag correlation between winter (JFM) precipitation in Iberian Peninsula and the anomalies of the FTLE for three different seasons through the period 1979-2008.

And,

The summer FTLE time series that show the higher correlation values with the winter precipitation cover approximately the area between 30°W and 0°W and between 25°N and 65°N. The size of the area chosen to correlate with the precipitation was varied within the North Atlantic Ocean without modifying significantly the results shown here.

p.5, l.1: What is the "IPNA region"?

We agree with the reviewer, IPNA was not defined in the text. IPNA (Iberian Peninsula North Atlantic).

p.5, l.33 and below: Please be specific in whether correlations are positive or negative.

We agree with the reviewer, and the text has been modified including the signs (+) or (-) to account for positive or negative anomalies.

Tab. 1: use capital letters for indicating calendar months

Modified.

p.6, l.9: SCA is not the third leading mode of WINTER SLP variability, but can be computed for all seasons (as every teleconnection index).

You are right, we have modified the text. The Scandinavia pattern (SCAND) consists of a primary circulation center over Scandinavia, with weaker centers of opposite sign over western Europe and eastern Russia/ western Mongolia. The Scandinavia pattern has been previously referred to as the Eurasia-1 pattern by Barnston and Livezey (1987). and other studies also show its influence on the Iberian Peninsula precipitation (deCastro et al., 2006; Casanueva et al., 2014).

In addition, the English could be further polished here and there, especially regarding the proper use of articles and (in just a few cases) the consistency of tenses.

We thank the Referee to point us this problem. We have revised the paper and hopefully the English style has been improved.

Please also note the supplement to this comment:
http://www.nonlin-processes-geophys-discuss.net/npg-2016-79/npg-2016-79-AC2-supplement.pdf
* * *
[Figure]

**Fig. 1.** Examples of FTLE intra and inter-annual variabilities

---

## Author Response (AR2)

We would like to thank the Referees for their valuable comments and critics that we tried to take into account in the revised version of the manuscript. Hopefully, all the major and minor corrections pointed out by the reviewers have been corrected now. A detailed answer follows below. We provide replies to the reviewer' comments in bold. Since most of the corrections suggested by the referees consisted of minor corrections to the text, all of them have been accepted and included in the new version of the text. Only those corrections that imply some discussion are commented below.

**Answers to Referee 1**

**The authors have partially addressed my concerns. I still believe the paper has interesting results that merit publication, but I do not think it is ready yet.**

**1) I still can not follow the reasoning behind the mechanism that links the summer FTLE and rainfall in winter. The authors say "In our opinion, large-scale tropospheric mixing drives summer SST anomalies that lead to changes in the next seasons storm tracks, and consequently changes in the location of action centers". However, it is not possible to deduce this from their results.**
**They show composites of different fields during summer for large + and - cases of FTLE anomalies. However, there is no causality implied there. SST lags atmospheric forcing for about 3 months, and thus the simultaneous maps shown can not be used to imply causality. Also, it is not obvious that midlatitude SST forces atmospheric circulation anomalies during winter that lead to rainfall.**
**An alternate possibility is that the FTLE in summer is a reflection of tropical SST anomalies, which may persist and then impact winter precipitation. Note that both winter rainfall and FTLE are correlated to SOI in JFM.**

Several previous studies have related the influence of North Atlantic SSTs on precipitation in different European areas, namely, in Sardinia, Italy (Delitala et al., 2000); in southwest England (Phillips and McGregor, 2002); in Iceland (Phillips and Thorpe, 2006); in Iberian Peninsula and Northern Africa (Rodriguez-Fonseca et al., 2006; Lorenzo et al., 2010). In all of them, a delay of several months for a maximum of correlation has been observed.

In this paper, we open a new possibility to understand this correlation. Our hypothesis is that summer FTLEs (large-scale tropospheric mixing occurring during summer) activate the anomalies of SST in the Atlantic Ocean modifying the next months mid-latitude atmospheric circulation. Changes in the circulation will lead to more or less precipitation in the west of the Iberian Peninsula.

The second possibility raised by the referee concerning the link between tropical SST anomalies and summer FTLEs could also be an explanation to the observed precipitation correlation. However, our studies suggest that the observed correlation between summer FTLES and the next winter SOI is only significant at 90%. We did not calculate the influence of winter SOI on next summer FTLEs.

Although out of the scope of this paper, the use of an atmospheric model to simulate the influence of atmospheric mixing onto the SST and its latter precipitation increase could be of great help.

References

- Cassou C, Deser C, Terray L, Hurrell JW, Drévillon M. 2004. Sea surface temperature conditions in the North Atlantic and their impact upon the atmospheric circulation in early winter. Journal of Climate **17**: 3349–3363.
- Delitala AMS, Cesari D, Chessa PA, Ward MN. 2000. Precipitation over Sardinia (Italy) during the 1946–1993 rainy seasons and associated large scale climate variations. International Journal of Climatology **20**: 519–541.

- Lorenzo MN, Iglesias I,Taboada JJ, Gómez-Gesteira M. 2010. Links between circulation weather types and teleconnection patterns and their influence on precipitation patterns in Galicia (NW Spain). International Journal of Climatology **30**: 980–990.
- Phillips ID, McGregor GR. 2002. The relationship between monthly and seasonal south-west England rainfall anomalies and concurrent North Atlantic sea surface temperatures. *International Journal of Climatology* **22**: 197–217.
- Phillips ID, Thorpe J. 2006. Icelandic precipitation-North Atlantic sea-surface temperature associations. International Journal of Climatology **26**: 1201–1221.
- Rodríguez-Fonseca B, Polo I, Serrano E, Castro M. 2006. Evaluation of the North Atlantic SST forcing on the European and Northern African winter climate. International Journal of Climatology **26**: 179–191.

**2) When describing Figure 3 (page 5) the authors propose a series of changes in synoptic activity without providing evidence for that. If they believe this is the right explanation they should at least say that this is an hypothesis, or speculation.**

We agree with the referee and we changed the explanation of Fig.3.

**3) In the response the authors mention that**
**"the changes in the values of positive phase and negative phase of FTLEs are small, the Wilcoxon rank sum test shows that these differences between positive and negative phase of FTLEs are significant. The same is found between the differences of the positive and negative anomalies of the other variables."**
**If this is certainly the case, they should explain in the manuscript the use of this test and mark in the figures the regions that are really significant. This is important given that the changes in FTLE are of the order of 3%.**

The Wilcoxon rank sum test is a nonparametric test for two populations.

Figure 3 shows the areas (pointed) in panels (a) and (b) where the differences between both FTLE phases are really significant, obtained from a two-sided Wilcoxon rank sum test. This test has been applied to both phases of the FTLE anomalies time series for each location shown in panels (a) and (b).

On the other hand, NOAA only provides spatial maps for the composites of anomalies of the other climatic variables, obtaining in all cases that both phases are significantly different. In this case we cannot show in which positions are significant or not. Otherwise, we should download the full climatic variables data sets and calculate the anomalies directly from the time series repeating the same procedure done for the FTLE.

Answers to Referee 2

**The authors have addressed some of the recommendations from my original review to improve the presentation of their work. However, I feel that some further revision is necessary before this work can be considered ready for publication.**

**My main concern is still the definition of seasons, which is non-standard. Despite the statements in the response letter, there is a climatological standard referring to DJF (JJA) as (boreal) winter (summer). If a climatologist reads the term "(boreal) summer/winter", they will most likely expect exactly these months to be considered. I accept that some authors may deviate from this general practice for probably good reason, but in this case, I strongly recommend clarifying throughout the manuscript that the three-month periods of JFM and JAS are used and NOT the common climatological winter and summer months. That is, I recommend replacing the terms "summer" and "winter" by JAS and JFM, respectively, throughout the manuscript to avoid any ambiguities.**

To avoid ambiguities we have tried to replace or include the acronyms JAS and JFM along the text.

**The second, yet minor point of criticism is a number of sentences where rephrasing is necessary for either scientific or grammatical reasons. I provide a list of these points below (page and line numbers refer to the final manuscript markup without tracked changes).**

We really want to acknowledge the referee for his/her detailed list of corrections that we are sure will improve the paper. All of them have been considered in the new version of the text. However, some of them are commented below.

**• Page 3, ll. 9-10: This sentence appears a bit misplaced at this point; better provide this information at the point where the correlations are discussed.**

The sentence has been removed from Section 2.1. and parts of it included when Fig.2 is described.

**• Page 4, ll. 10-14: Please detail the meaning of the terms "mean" and "total mean" (i.e., "mean" seems to denote the spatial mean at a given time step, and total mean the average FTLE over all time steps?). Also, the phrase "two time series (positive and negative phases)" is not clearly understandable as such.**

*Total mean* refers to the climatological period 1981-2010 used for the anomaly plots by the Earth System Laboratory (NOAA). While *mean* corresponds to the mean of years with positive/negative (above/below the median) summer FTLE. Then, two time series (positive and negative phases) of these seasonal composites were calculated for years with positive/negative summer FTLE anomalies. We have corrected the last paragraph in Section 2.2 to better describe these terms.

**• Table 1: Please clarify if the seasons (rows) correspond to the same year as "winter"/"summer" (should be "JFM"/"JAS") or the preceding/following.**

We agree with the referee that it was not clear the timing of the seasons. To avoid that, we have included the symbols +1, 0 and -1 to indicate that the correlation is with the next, same or previous year season. Thus, for example, summer (JAS) FTLE correlate with next year AMJ NAO with a coefficient equal to 0.34, and winter (JFM) rainfall correlates with previous year OND SCA with a coefficient equal to 0.36.

**• Fig. 3, caption: What do you mean by "global summer FTLE mean" – surely not the summer FTLE mean around the globe. Moreover, when you select years with positive/negative FTLE**

**anomalies – do you mean all years where the mean FTLE is above/below its global mean, or just such where this deviation is particularly strong (as one would commonly use when considering "climatological composites")?**

We agree with the referee that this sentence was not clear. We have deleted the word "global" as it has no meaning at all in this context. The meaning of summer FTLE mean has been explained at the end of Section 2.2 in agreement with your question answered above.

Following the recommendations of the reviewer we have rephrased the paragraphs or added some sentences as suggested. All of these corrections have been included in the new version of the text. However, some of them are commented below.

**First of all, I think that include some references in several parts of the text can improve the quality of the manuscript. I suggest include references on the next points:**
**\* Page 1: When the authors write: "However, the relationship between NAO and ENSO and the European variability is nonstationary; that is, the strength of the correlation between these two teleconnections and climate anomalies has changed over time."**

These references are already in the paper al the end of the next sentence where we discussed the non-stationary of these two modes. (Vicente-Serrano and López-Moreno, 2008; Rodríguez-Fonseca et al., 2016).

**\* Page 2: On the affirmation: "Changes in mid-latitude circulation can strongly affect the weather events", I suggest to add a reference and also a region where this changes can be observed, referring the type of weather events that are affected.**

We added the suggestion of the referee in the paper;
- Screen, J. A., and I. Simmonds, Amplified mid-latitude planetary waves favour particular regional weather extremes. Nat. Climate Change **4**, 704–709, 2014.
- Marshall, J. et al. North Atlantic climate variability: phenomena, impacts and mechanisms. Int. J. Climatol. **21**, 1863–1898, 2001.

**\* Page 2 and 3: Please include a reference or web for the IB02, ERSST, ICOADS, NCEP reanalysis and CPC (related with teleconnection indices) databases.**

We have added new references and a new Table with the web sites.

**Other comments and suggestions are the next ones:**

**Page 2:**
**The authors include some works that link the Iberian precipitation with other variables. The paragraph is the next one:**
**"Previous works have shown a possible link between the Iberian precipitation and other variables like summer Sea Surface Temperature (SST) anomalies over the north Atlantic basin (Rodríguez-Fonseca and deCastro, 2002; Lorenzo et al., 2010; Hatzaki et al., 2015), other teleconnection patterns (deCastro et al., 2006; Casanueva et al., 2014) or the Euroasian snow cover in autumn (Brands et al., 2014). The storm track activity has been related to the occurrence of extreme events (Lehmann and Coumou, 2015). Changes in mid-latitude circulation can strongly affect the weather events."**
**Please, can you clarify how the Iberian precipitation can be linked with the Euroasian snow cover in autumn, and also where the storm track activity was considered and where the occurrence of extreme events was analysed?**

The main storm-track activity occurs in mid-latitudes. The paper by Brands et al. (2014) describes the correlation among the Iberian precipitation and the Euroasian snow cover in autumn.

**I also suggest to include, in a short way, the main methods used to calculate the FTLE in the next paragraph: "Our goal in this study is to characterize the rainfall patterns in the Iberian Peninsula as a function of the large-scale tropospheric mixing over the Atlantic ocean. To that end, we have calculated a climatology of FTLE for the period 1979-2008. The obtained**

**time series was then correlated with the precipitation over the Iberian Peninsula. Finally, we discuss the obtained results by considering their relationship to the main modes of circulation variability."**

The sentence "*To that end, we have calculated a climatology of FTLE using finite-difference approximation to the deformation gradient for the period 1979-2008.*" has been included.

**Page 2 and 3:**

**The authors said that they considered "Yearly anomalies of the SST, geopotential at 500 hPa, sea level pressure (SLP) and wind speed at 200 hPa and 850 hPa for the same period have been used. The SST anomalies have been derived from The Extended Reconstructed Sea Surface Temperature (ERSST) dataset which is a global monthly sea surface temperature dataset derived from the International Comprehensive Ocean-Atmosphere Dataset (ICOADS). It has been derived on a 2 x 2 grid with spatial completeness enhanced using statistical methods. This monthly analysis begins in January 1854 continuing nowadays. The newest version of ERSST, version 4, is based on optimally tuned parameters using the latest datasets and improved analysis methods. The geopotential, SLP and wind speed anomalies have been obtained from the National Center of Environmental Prediction (NCEP) reanalysis with a spatial resolution of 2.5 x 2.5. The climatology used for the anomaly plots is for the 1981-2010 period." (page 2 and 3). Nevertheless, in the page 4 they said that "Seasonal composites (averages) of the anomalies (mean - total mean) of the SST, geopotential height and wind speed were obtained from NCEP for the same period." Please, can you clarify which kind of data is used and with which temporality? Also the authors said that the performed a climatology for the anomaly plots using the 1981-2010 period. Why this period was selected and not the 1979-2008 period that is the one chosen for the entirely work?**

We agree with the reviewer that the period selection could be confused so we added at the end of section 2.1 the following sentence;

*The climatology used for the anomaly plots is for the 1981-2010 period. This period is considered to be the standard period for climatological studies of anomalies.*

**Please, considered to place the sentence: "The monthly indices were averaged for the seasons JFM, AMJ, JAS and OND from 1979 to 2008. The most representative atmospheric patterns for the Northern Hemisphere were considered in order to analyse their influence on precipitation for the Iberian Peninsula and with the summer FTLEs." at the beginning of its paragraph on this way: "The monthly indices were averaged for the seasons JFM, AMJ, JAS and OND from 1979 to 2008. The most representative atmospheric patterns for the Northern Hemisphere were considered in order to analyze their influence on precipitation for the Iberian Peninsula and with the summer FTLEs. The teleconnection indices NAO, SCA (Scandinavia pattern), EA (East Atlantic pattern), EA/WR (East Atlantic/ West Russia pattern), POL (The Polar/ Eurasia pattern), SOI (Southern Oscillation Index), PNA (Pacific-North American Pattern) and the atmospheric mode AO (Artic Oscillation) were obtained from the Climate Prediction Center (CPC) at NCEP at monthly time scales."**

We consider that it was correctly written as monthly indices refer to the teleconnection indices previously described.

**Why the authors considered an initial separation of 0.35º in the distribution of the particles?**

The FTLE measure the deformation of an approximation to a continuum material surface. The model resolution is 0.7ºx0.7º, so at least the minimum resolution considered should be 0.7º since all

points should be forced by the wind field at starting times in order to have a smooth deformation field. To that end, we have considered a four times higher resolution to solve this item without compromising so much the computational cost.

**Page 4:**
**It will be interesting to comment that the FTLE were initially calculated globally, and then an area of the Eastern Atlantic was extracted to perform the study. Please, considered to include a square in the Figure 1 that represent the Eastern Atlantic area used to extract the FTLE results (between 30ºW and 0ºW and between 25ºN and 65ºN).**

Included.